# Optimal Strategy to Tackle a 2D Numerical Analysis of Non-Uniform Flow over Artificial Dune Regions: A Comparison with Bibliography Experimental Results

**Jungkyu Ahn [1], Jaelyong Lee [1] and Sung Won Park [2],***

[1]  Department of Civil and Environmental Engineering, Incheon National University, Incheon 22012, Korea; ahnjk@inu.ac.kr (J.A.); dlwofydsla1@naver.com (J.L.)
[2]  Department of Data-centric Problem-Solving Research, Korea Institute of Science and Technology Information, Daejeon 34141, Korea
*   Correspondence: swpark@kisti.re.kr; Tel.: +82-42-869-1624

**Abstract:** Flow simulation over a dune requires the proper input of roughness coefficients. This study analyzed a numerical simulation of open-channel turbulent flow over two-dimensional fixed dunes to reveal the effect of roughness on the dune bottom, and to determine the optimized combination of the turbulence scheme and the roughness height formula. The most appropriate roughness values and turbulence models were applied using Reynolds-averaged Navier–Stokes models. Seven methods were chosen to estimate the bed roughness properties at the inlet boundary section. The results of all cases calculated with the OpenFOAM toolbox were compared with laboratory experimental data for model validation. The performances of all bed roughness variations were evaluated according to the stream-wise and depth-wise directions with nondimensional values. Consequently, it was revealed that the combination of bottom roughness length scale at the inlet boundary and the $k$-$\omega$ shear-stress transport (SST) model was the most suitable for the flow separation zone and turbulent properties near the channel bottom.

**Keywords:** bed roughness; open-channel flow; OpenFOAM; Reynolds-averaged Navier–Stokes model; turbulence model; two-dimensional dune

---

## 1. Introduction

Bed forms such as ripples and dunes in an alluvial river change naturally with the interaction between the river and sediment. In particular, because of scour around hydraulic structures such as piers and dams, bottom protection of weirs is considered to be an important phenomenon of sediment transport, since the excessive loss of bed materials can have severe effects and cause damage to structures [1–4]. In general, turbulent flow with arbitrary flow characteristics is considered to be the predominant cause of scouring. This process is too complicated to predict because it is a result of the relationships among the water flow around structures and the irregular bed properties in natural rivers. Analysis of the characteristics of turbulent flow in rivers requires interaction between the flow and the channel bed, and one of the most important processes for the numerical analysis is the determination of the bed roughness.

Most numerical models of local scour, sediment transport, and deposition separately use an uncoupled scheme, turbulence, and sediment transport [5]. The flow computation is conducted before the sediment computation. Then, the bed shape is updated and processed to the next time step. In the present study, we analyzed turbulent structures in flow over fixed dunes to find the best method for

---

turbulent computation. Among the various irregular bed shapes in alluvial rivers, the dune is one of the most interesting in context of understanding its interaction. Numerous experimental approaches on this topic have been carried out [6–16]. Particularly, it has been revealed that dunes were formed by multiple mechanisms resulting from the complex combination of bedform, grain motions, and flow dynamics that evolve as a bed develops [17]. Five major regional flow characteristics are summarized in Figure 1 based on previous researches [15,17]. Therefore, this study focused on the effects of bed roughness on turbulent flow, which is one of the factors influencing flow in natural streams.

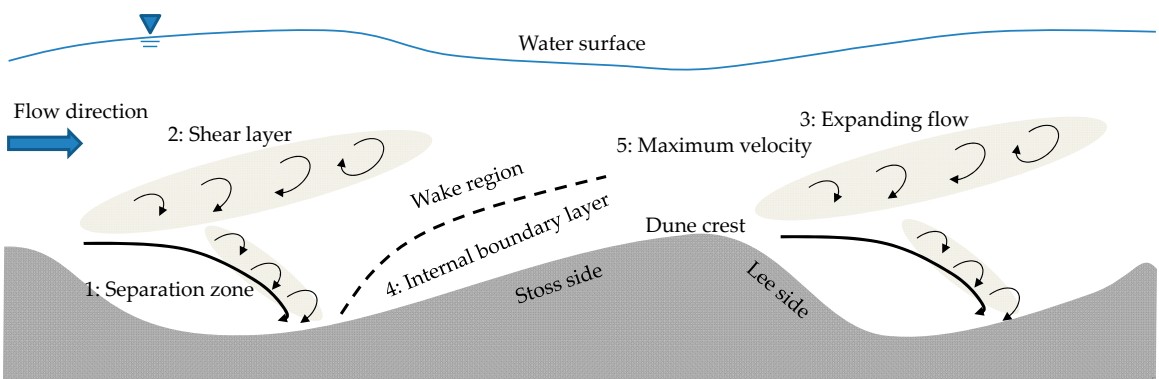

**Figure 1.** Schematic diagram of principal regions of flow over the dunes [15,17].

Physical and numerical approaches can be applied to investigate bed roughness and its effects. The physical approach requires no parameters to start modeling. However, it is very difficult to maintain constant and stable conditions during modeling, especially for large scale models. In addition, measuring error is inevitable. A considerable disadvantage is the scale effect. Therefore, numerical simulation is widely applied because it minimizes the scaling effects and facilities [18–25]. Although adequate results can be obtained with this approach, there are many limitations owing to the combination of complex properties of the turbulent flow. Therefore, the appropriate combination of the model parameters and the turbulent modeling technique is essential.

Most numerical simulations of turbulent flow with various turbulence schemes have been studied to solve the Navier–Stokes (NS) equation to capture the effect of turbulence motion. Direct numerical simulation (DNS) is a simulation in computational fluid dynamics (CFD) for solving NS equations of turbulent flow without additional turbulence schemes. This method is the most accurate for modeling flow characteristics. However, the computational cost of DNS is high because all spatial scales of the turbulence must be resolved in a mesh from the minimum dissipated scale, and it is difficult to simulate for large, complex applications [26–28]; nonetheless, turbulence modeling is conducted in practical engineering applications. To overcome the diseconomy of the DNS method in terms of computational time and effort, large eddy simulation (LES) methods with lower computational costs and the Reynolds-averaged Navier–Stokes (RANS) methods based on averaging the flow equations yielding the Reynolds-Averaged Navier–Stokes (RANS) equations were applied with accurate flow simulations [28,29]. The closure problem was solved with transport equations to reproduce the behavior of turbulent flow, and then the turbulence scales were related to a turbulent viscosity. Two-equation models provided a full description of turbulence in terms of length and time scales; thus, they facilitated the reproduction of various flow patterns [26,28].

In this study, we used the turbulence model based on the RANS equations to simulate the numerical modeling for turbulence flow over a fixed dune, and to reduce the computational time by solving the mean velocity and effects of the temporal fluctuation. Numerous numerical studies have provided reasonable results for turbulent flow on sandy dunes using various RANS models [2,23,30–36]. Considering the previous studies, we selected *k-ε* and *k-ω* shear-stress transport (SST) turbulence models from the RANS models. In addition, a two-dimensional (stream-wise and depth-wise direction) model was applied for flow and turbulence analysis over fixed dunes because these have been

considered to be the dominant factors of the maximum scour depth in the design criteria of hydraulic structures in previous researches [1,4,37–40].

In addition, bed roughness, which depends on the size of the bed materials, is one of the most important factors in open-channel flow. This property is presented through various suggestions based on previously suggested experimental data and is expressed as the grain size of the bed material. In the general type of numerical modeling, properties of bed materials are chosen according to empirical formula. Representatively, several identical dunes were studied to identify the effect of the dune wall roughness [35] and dimensions by using the standard $k$-$\varepsilon$ model and the Spalart–Allmaras model [41] with one-equation models and by applying the Colebrook–White formula [42] to set the roughness value of the wall. In addition, previous studies [8,43] have shown the results of various hydrodynamic factors, i.e., the time-averaged velocity, eddy viscosity, free-surface elevation, wall stress, wall pressure, friction, and form resistance data, to determine a suitable formula for bed roughness. A numerical analysis was conducted to investigate the form drag of several idealized dune configurations and to provide various flood wave models by using the Nikuradse roughness parameter from previous research [44,45]. Their predictions of the velocity profiles and flow separation zones were in general agreement with the experimental data from previous research [46]; however, only the standard $k$-$\varepsilon$ model was used. Therefore, the objective of this study was to apply several combinations of empirical formula for the bed roughness in the numerical approaches for the fixed dune, and to determine the most suitable formula. Six experimental formula were used for turbulent distribution on the bed of the artificial dune using the data from previous research [6].

The physical experimental data of dune profiles and conditions were obtained from the previous research [6]; these data have been widely used and proven to be accurate in previous studies [2,23,32,35,36,47,48]. We employed the OpenFOAM (The OpenFOAM Foundation Ltd. London, United Kingdom) toolbox, an open-source CFD platform based on a cell-centered finite volume method. The simulated results of flow velocity and turbulent kinetic energy were obtained, and the optimal best combinations were determined by comparing the results with those of physical modeling [6]. The goodness-of-fit was quantified by root mean square error (RMSE).

## 2. Numerical Method

The OpenFOAM toolbox is a free CFD program developed by OpenCFD based on C++, a collection of solvers capable of simulating the flow of various fluids, and open code software. This toolbox is basically a three-dimensional model that solves the partial differential equation based on the finite-volume method and numerically presents the governing equation of fluid motion. The governing equation for incompressible flow is the RANS equation using the Reynolds-averaged theorem. The Reynolds stresses are modeled based on the eddy viscosity. The continuity equation and the momentum equation in vector form are as follows:

$$\nabla \cdot \vec{u} = 0 \tag{1}$$

$$\frac{\partial \vec{u}}{\partial t} + (\vec{u} \cdot \nabla)\vec{u} = -p + \nabla \cdot \left[ (\nu + \nu_T)(\nabla \vec{u} + \nabla \vec{u^T}) \right] \tag{2}$$

where $\vec{u}$ is the velocity vector, and upper $T$ denotes the transposed gradient, $p$ is the pressure, $\nu$ is the kinematic viscosity coefficient, and $\nu_T$ is the turbulent kinematic eddy viscosity coefficient which is supposed to simulate the effect of unsolved velocity fluctuations.

In the OpenFOAM toolbox, the standard $k$-$\varepsilon$ turbulence model [49] is one of the most commonly used turbulence models to calculate flow properties for turbulent flow. It describes turbulence by means of two partial differential equations (PDEs) for the following two variables: turbulent kinetic energy (TKE), $k$ ($= 0.5 \sqrt{\overline{u'^2} + \overline{v'^2} + \overline{w'^2}}$, where $u'$, $v'$, and $w'$ denote the three-dimensional velocity fluctuation from the time-averaged velocity data); and the turbulent dissipation, $\varepsilon$. This model is robust, economical, and relatively accurate for many cases [40]. It is designed as a high Reynolds number

model and provides a relatively reasonable formula, especially in fully developed turbulent flow; it is simple to implement and performs relatively well for boundary flow. However, the standard model fails to properly predict flows close to the wall region where the Reynolds number is low, because the eddy viscosity is overestimated by the two equations in this case [44,50]. Therefore, the wall function was applied to improve the calculation accuracy near the wall; it was applied to the first cell of the wall boundary, thus applying a constant roughness to these walls. The turbulent viscosity can be determined by two convention-diffusion-reaction equations for computation, presented as follows:

$$\frac{\partial k}{\partial t} + \nabla \cdot (\vec{u} k) - \nabla \cdot (\nu + \nu_T) \nabla k = P_k - \frac{\varepsilon}{k} \tag{3}$$

$$\frac{\partial \varepsilon}{\partial t} + \nabla \cdot (\vec{u} \varepsilon) - \nabla \cdot (\nu + \nu_T) \nabla \varepsilon = (C_1 P_k - C_2) \frac{\varepsilon}{k} \tag{4}$$

where $P_k$ is the turbulent viscosity production due to viscosity. In addition, $\nu_T$ is defined as:

$$\nu_T = \frac{C_\mu \sqrt{k}}{\varepsilon} \tag{5}$$

where $C_1$, $C_2$, and $C_\mu$ are the empirical constants with values of 1.44, 1.92, and 0.09, respectively.

The $k$-$\omega$ SST model is a two-equation model developed by [50] and is similar to the $k$-$\omega$ model [51]. The $k$-$\omega$ SST model effectively combines the formulations of the $k$-$\varepsilon$ and $k$-$\omega$ models. The model assumes that the turbulent viscosity is related to values of $k$ and $\omega$. The $k$-$\varepsilon$ model is similar to the $k$-$\omega$ model but it is more accurate for wall effect treatment. However, the SST model avoids the problem of the $k$-$\omega$ model's sensitivity to inlet properties in freestream, by switching the formula of both models to each other in freestream [50]. Therefore, the turbulence characteristics near the wall and the freestream region can be predicted appropriately, and in particular, the flow separation zone can be predicted appropriately near the upstream of the flow [50]. The two equations for the computation of $k$ and $\omega$ are as follows:

$$\frac{\partial k}{\partial t} + \nabla \cdot (\vec{u} k) - \nabla \cdot (\nu + \sigma_k \nu_T) \nabla k = P_{\omega k} - \beta^* k \omega \tag{6}$$

$$\frac{\partial \omega}{\partial t} + \nabla \cdot (\vec{u} \omega) - \nabla \cdot (\nu + \sigma_\omega \nu_T) \nabla \omega = \alpha S^2 - \beta \omega^2 - (F_1 - 1) CD_{k\omega} \tag{7}$$

$$F_1 = \tanh \left\{ \min \left[ \max \left( \frac{\sqrt{k}}{\beta^* \omega z}, \frac{500\nu}{z^2 \omega} \right), \frac{4 \sigma_{\omega 2} k}{CD_{k\omega} z^2} \right]^4 \right\} \tag{8}$$

$$CD_{k\omega} = \max \left( 2 \sigma_{\omega 2} \frac{1}{\omega} \frac{\partial k}{\partial x_i} \frac{\partial \omega}{\partial x_i}, 10^{-10} \right) \tag{9}$$

$$\nu_T = \frac{a_1 k}{\max(a_1 \omega, S F_2)} \tag{10}$$

$$F_2 = \tanh \left\{ \left[ \max \left( \frac{2 \sqrt{k}}{\beta^* \omega z}, \frac{500\nu}{z^2 \omega} \right) \right]^2 \right\} \tag{11}$$

$$\alpha_i = \alpha_1 F_1 + \alpha_2 (1 - F_1) \tag{12}$$

$$\beta_i = \beta_1 F_1 + \beta_2 (1 - F_1) \tag{13}$$

where $F_1$ and $F_2$ are blending functions; $CD_{k\omega}$ and $P_{\omega k}$ are the limited production functions of the turbulent viscosity; $\omega$ is the rate of dissipation of the eddies; $\alpha$, $\beta$, and $\sigma$ coefficients [52] are $\alpha_1 = 0.5532$, $\alpha_2 = 0.4403$, $\beta_1 = 0.075$, $\beta_2 = 0.0828$, $\beta^* = 0.09$, $\sigma_k = 0.8503$, $\sigma_{\omega 1} = 0.5$, and $\sigma_{\omega 2} = 0.8561$; $a_1 = 0.31$; and $S$ is the absolute value of the vorticity [53]. A detailed description of the model parameters is given in [50,52].

## 3. Model Setup

The numerical model was set up to replicate the laboratory experiments [6], which analyzed the flow characteristics along a flume in two-dimensional fixed sand dunes. Laboratory experiments were carried out with an artificial dune model of 1.6 m length, 0.08 m height, and 1.5 m width with a glued-bed median grain size of $D_{50} = 1.6$ mm. The flow properties in the experiments were measured in single parts of the dunes. The results of the experiments were used to validate numerous numerical researches [22,23,36,47]. One of the experimental cases [6] was selected to evaluate the numerical calculation in this study. The conditions of the selected cases are shown in Table 1.

**Table 1.** Dimensions of the artificial dune and experimental conditions.

| $L$ (m) | $h$ (m) | $b$ (m) | $D_{50}$ ($10^{-3}$ m) | $T$ (°C) |
|---|---|---|---|---|
| 1.6 | 0.08 | 1.5 | 1.6 | 18 |

The computational domain in this research is a two-dimensional single dune, as in Figure 2. To improve the simulation results in the vicinity of the bottom, a mesh that became gradually finer toward the bottom was generated, as shown in Figure 2. Experimental results were validated to show grid size convergence using the $k$-$\varepsilon$ model with three different sizes of $\Delta z$ (first node from the wall) [22]. Therefore, we added a 1 m stabilization zone, where the number of meshes was 11,000 (200 in the $x$ direction and 55 in the $y$ direction) in front of the inlet boundary. For the main body of the simulation, 17,600 meshes were setup with 320 and 55 meshes in the $x$ and $y$ directions, respectively. The boundary conditions for seven cases of different bed roughness heights were determined, as shown in Table 2.

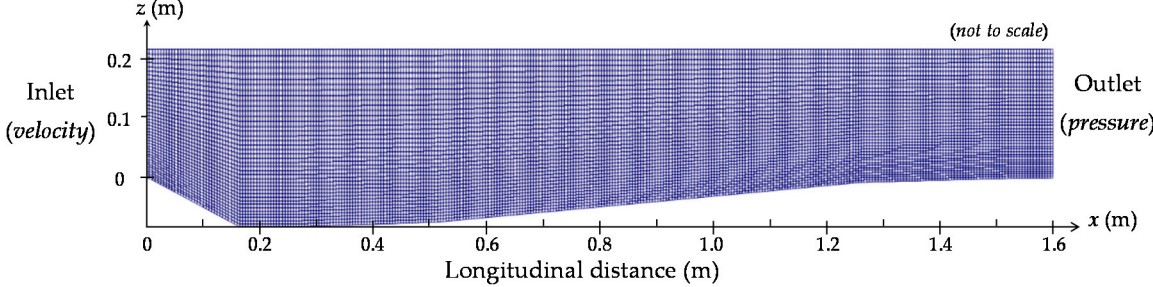

**Figure 2.** Computational domain of two-dimensional open-channel flow over single dune.

**Table 2.** Boundary conditions of turbulent models.

| $h_0$ (m) | $u_0$ (m s$^{-1}$) | $k$ ($10^{-4}$ m$^2$ s$^{-2}$) | $\varepsilon$ ($10^{-4}$ m$^2$ s$^{-3}$) | $\omega$ (s$^{-1}$) | $\Delta z$ (mm) |
|---|---|---|---|---|---|
| 0.213 | 0.466 | 8.14 | 2.56 | 3.49 | 0.15 |

The vertical profile of the stream-wise flow velocity is controlled by the water depth, the slope, and the roughness of the wall [45]. In particular, wall roughness affects the bottom shear stress as well as the logarithmic velocity profile in uniform flow. The effect of the bed roughness is more dominant in the recirculation zone, which has been referred to as the separation zone [54,55]. The bed roughness for flow in natural rivers depends on the total induced resistance. Bed roughness can also be expressed by the roughness height. Therefore, it is necessary to set the appropriate roughness height on the wall to obtain a more accurate numerical result. We selected and applied seven experimental cases from empirical formula used mainly for open-channel cases. Values of roughness height used in the laboratory experiments [6] and values from sand grain are summarized in Table 3.

**Table 3.** Roughness values of cases.

| Case No. | Case 1 | Case 2 | Case 3 | Case 4 | Case 5 | Case 6 | Case 7 |
|---|---|---|---|---|---|---|---|
| Formula | $D_{50}$ | $1.6D_{50}$ | $3D_{90}$ | $3.5D_{84}$ | $5.2D_{65}$ | $5.1D_{84}$ | $6.8D_{50}$ |
| $k_s$ (mm) | 1.6 | 2.5 | 5.55 | 6.3 | 8.84 | 9.18 | 10.88 |

## 4. Results and Discussion

### 4.1. Flow Velocity and Turbulence Distribution

Two turbulent modeling schemes ($k$-$\varepsilon$ and $k$-$\omega$ SST) and seven roughness heights were considered for the simulations to find the most suitable method for determining the bed roughness properties using the turbulence model. The simulated results were compared to measured data from [6]. The numerical results with roughness height 2.5 mm (Case 2) set by the numerical modeling are shown in Figures 3–5. The profiles of the dimensionless flow velocity of the flow direction ($u_x/u_0$) and vertical direction ($u_z/u_0$) as well as the dimensionless TKE ($k/u_0^2$) were plotted with respect to the dimensionless vertical position ($z/h_0$). Overall, the profiles of the numerical results of both $k$-$\varepsilon$ and $k$-$\omega$ SST models are in good agreement with the measured data, especially underneath $z/h_0 = 0$. However, the velocity profiles of the $k$-$\omega$ SST modeling results are in better agreement than those of $k$-$\varepsilon$ modeling results near the bottom and the separation zone from $x = 0$ m to $x = 0.3$ m. The TKE values are defined by the mean value of kinetic energy related to the turbulence, and results of both models showed remarkable trends of turbulent intensity along the dune crest line. Numerical results of the $k$-$\omega$ SST model were slightly better than those of $k$-$\varepsilon$ modeling overall, as analyzed by previous researches [23]. Thus, the $k$-$\varepsilon$ model had a limit of not being suitable for simulating the flow in the vicinity of the channel bottom, because the model was developed for models with large Reynolds numbers to solve the flow with relatively higher turbulent regions. Therefore, wall function was applied to calculate the flow and turbulence near the bottom and this function was applied to the first cell from the bottom which was able to treat the roughness effect [23,40].

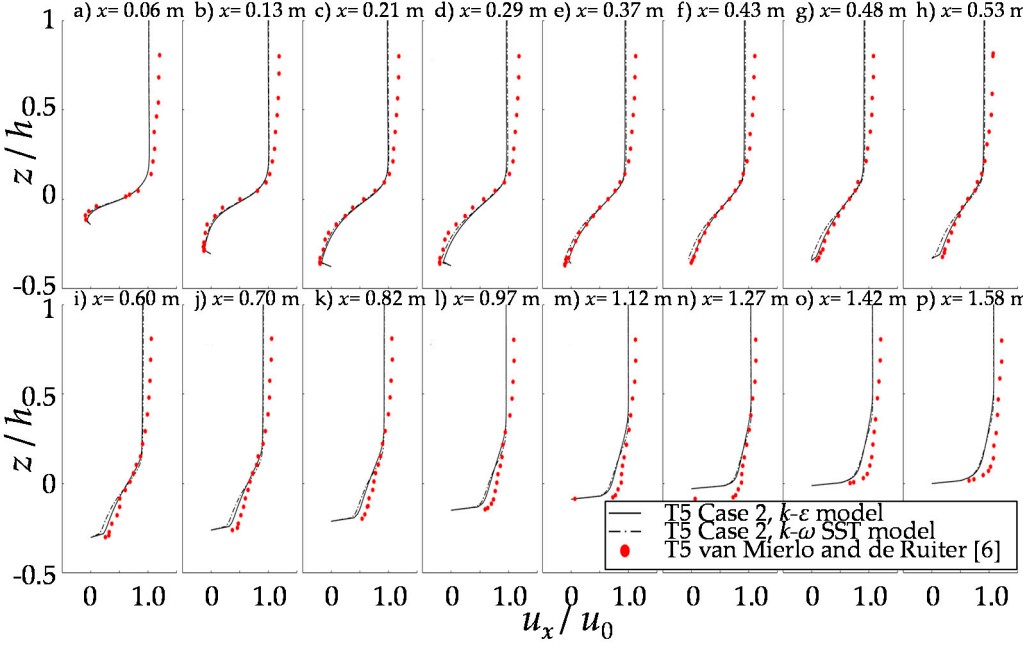

**Figure 3.** Comparison of different turbulence schemes in nondimensional stream-wise velocity with experimental data for Case 2 ($k_s = 1.6D_{50}$).

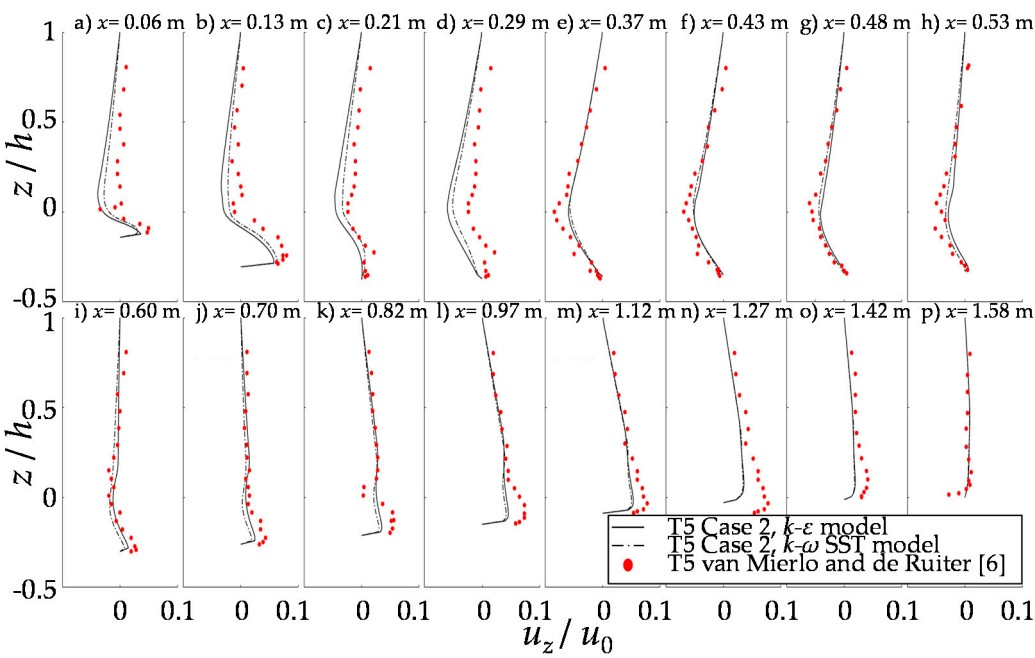

**Figure 4.** Comparison of different turbulence schemes in nondimensional depth-wise velocity with experimental data for Case 2 ($k_s = 1.6D_{50}$).

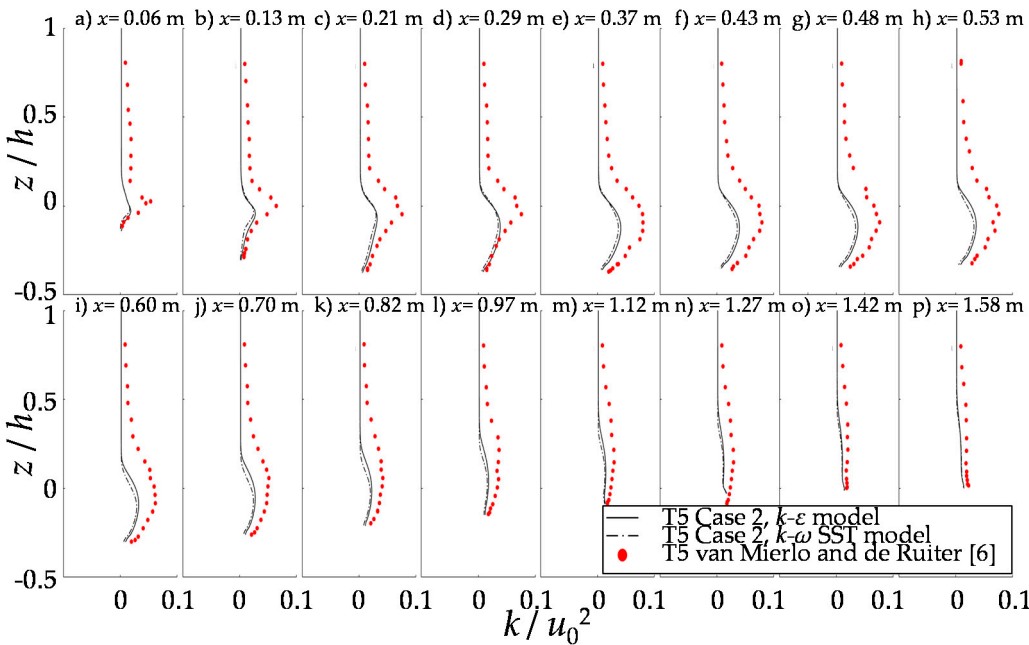

**Figure 5.** Comparison of different turbulence schemes in nondimensional turbulent kinetic energy (TKE) with experimental data for Case 2 ($k_s = 1.6D_{50}$).

### 4.2. Comparison with Experiments

Numerical results of all cases were quantified using RMSE for analysis to determine more suitable case roughness values because there was no remarkable variation in the profiles from those of Case 2, both by $k$-$\varepsilon$ and $k$-$\omega$ SST. The results of the RMSE were expressed about two regions, i.e., from the model bottom to the free surface, and the bottom to the dune crest depth. The RMSE was calculated as follows:

$$\text{RMSE} = \sqrt{(u_{\text{mod}} - u_{\text{exp}})^2} \tag{14}$$

where $u_{mod}$ is the numerical result and $u_{exp}$ is measured data from the experiment. We considered all vertical measured values as well as the lower part of the water depth deeper than $z/h_0 = 0$ to focus on the roughness effect near the separation zone by comparing the RMSE values. The effect of bed roughness variation is predominant near the wall region from the bottom to approximately 80% of the dune height [54]. Measuring points with more than three measurements were selected in this analysis. Therefore, there were 16 measuring points in the experiment, 13 of which were used to represent the RMSE results (points at $x$ = 1.27, 1.42, and 1.58 m were excluded). The distributions of two RMSE values are compared in Figures 6–8.

Overall, the averaged RMSE of the $u_x$ for the whole water depth (Figure 6) was stable, whereas the partially averaged results (Figure 6c,d) exhibited large differences upstream and downstream in the lower part (from initial bed elevation to the deepest point) of $z/h_0$ where the flow velocity fluctuated remarkably. In addition, the RMSE values of Case 1 and Case 2, which had relatively small roughness height values, showed a similar tendency. Case 3 to Case 7, which had greater roughness height values, exhibited identical patterns (Figure 6). The RMSE results of the upstream part are also less than those at the downstream. All water-depth averaged RMSE values were generally similar to those at the downstream direction (Figure 6a,b). However, the RMSE values in the lower part, from Case 3 to Case 7, increased at the downstream part of the dune (Figure 6c,d). This pattern is consistent with the fact that the dune geometry is a dominant factor of the entire depth of flow and that the effect of the bed roughness is predominant in the separation [54,55]. The RMSE values of both turbulence models in the lower part are smaller than the RMSE values of the entire water depth, especially at the upstream, and the RMSE of the $k$-$\omega$ SST model is slightly smaller than that of the $k$-$\varepsilon$ model. In particular, the smallest value of RMSE with the $k$-$\omega$ SST model is smaller than that of the $k$-$\varepsilon$ model, and the point changed from $x$ = 0.43 m in the $k$-$\varepsilon$ model to $x$ = 0.29 m in the $k$-$\omega$ SST. model (Figure 6c,d). The fully water-depth averaged values of RMSE with both turbulence models were estimated to be 0.0385–0.0630 and 0.0392–0.0613 m/s, respectively. Moreover, the values in the lower part were estimated to be 0.003–0.1416 m/s for the $k$-$\varepsilon$ model and 0.0072–0.0152 m/s for the $k$-$\omega$ SST model. The smallest values of RMSE in the lower vicinity of the dune upstream were 0.0111, 0.0124, and 0.0135 m/s in Case 3 of $k$-$\omega$ SST, at $x$ = 0.21 m, respectively. The suggested combination of turbulence model and initial condition for the roughness height in the inlet part were evaluated for each focus of analysis such as overall flow and turbulence distributions, as well as regional backwater trend at the downstream of the dune inlet.

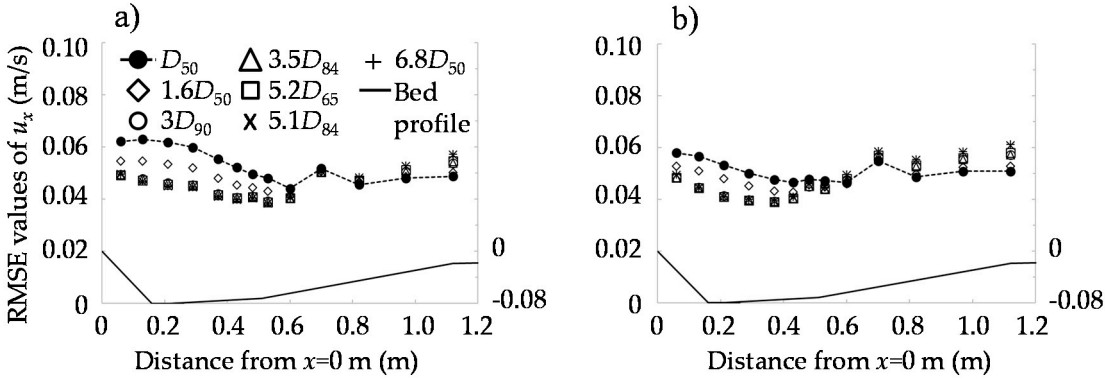

**Figure 6.** *Cont.*

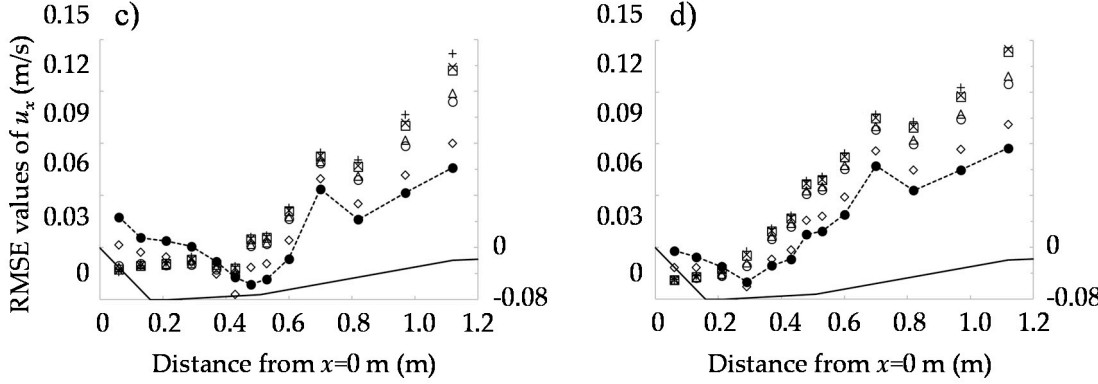

**Figure 6.** Root mean square error (RMSE) values of flow velocity in stream-wise direction with various bottom roughness height formulas: (**a**) Fully depth-averaged ($0 < z/h_0 < 1$) RMSE with $k$-$\varepsilon$ model; (**b**) Fully depth-averaged with $k$-$\omega$ shear-stress transport (SST) model; (**c**) Partially depth-averaged ($0 < z/h_0 < 0.15$) $k$-$\varepsilon$ model for the lower part; (**d**) Partially depth-averaged ($0 < z/h_0 < 0.15$) $k$-$\omega$ SST model.

The longitudinal distribution of the RMSE values of $u_z$ are plotted in Figure 7. The overall RMSE trends are similar for both turbulence models. Higher roughness value cases (Cases 3–7) showed better fit than lower cases (Cases 1 and 2). For the $k$-$\varepsilon$ model (Figure 7a,c), the RMSE values at the upstream part up to $x = 0.13$ m were larger than at the downstream part. The depth-averaged RMSE values of the two turbulence models were estimated to be 0.0028–0.0148 and 0.0029–0.0120 m/s, respectively, and the RMSE values in the lower part near the upstream up to $x = 0.21$ m were estimated to be 0.0078–0.0156 and 0.0058–0.0114 m/s for the $k$-$\varepsilon$ and $k$-$\omega$ SST models, respectively.

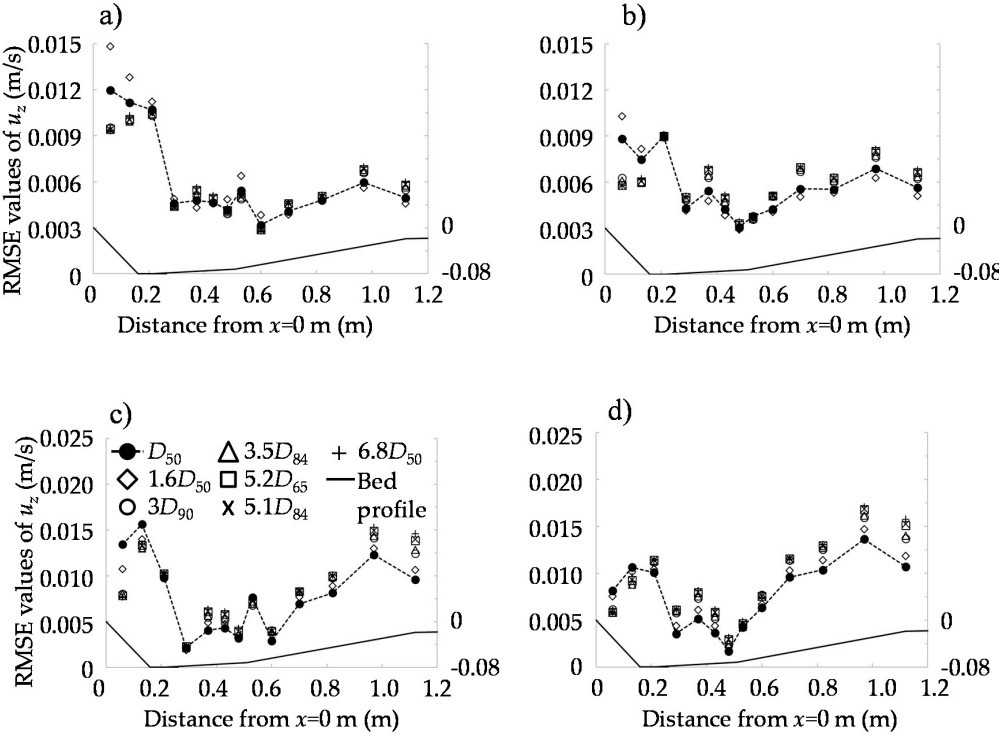

**Figure 7.** RMSE of flow velocity in depth-wise direction with various bottom roughness height formulas: (**a**) Fully depth-averaged ($0 < z/h_0 < 1$) RMSE with $k$-$\varepsilon$ model; (**b**) Fully depth-averaged with $k$-$\omega$ SST model; (**c**) Partially depth-averaged ($0 < z/h_0 < 0.15$) $k$-$\varepsilon$ model for the lower part; (**d**) Partially depth-averaged ($0 < z/h_0 < 0.15$) $k$-$\omega$ SST model.

The RMSE values of the TKE results about all cases are shown in Figure 8. The RMSE values were estimated to be relatively large downstream of the separation zone. These results were reflected due to the high turbulence in the wake region arising directly behind the flow separation zone [15] based on a comparison with the experimental data in Figure 3. Larger roughness cases (Cases 3–7) have substantially lower depth-averaged RMSE values than the smaller roughness cases for the entire application area, as shown in Figure 8a,b. The RMSE values in the lower part (Figure 8c,d) were similar for all cases to the wake region, and the RMSE values of the larger roughness cases had a better fit at the downstream. Theoretically, the $k$-$\varepsilon$ model should have the disadvantage of inaccurate flow properties for simulations near the wall region because this model was designed for large Reynolds number cases to solve fully developed turbulent flow. However, the results from the two models showed minor differences because the wall function was applied to overcome the limitation and to calculate the roughness effect, as mentioned previously [23]. In this study, the RMSE values of the TKE in the lower part were difficult to use for comparison of various roughness cases, as shown in Figure 8c,d. The characteristics of TKE are not representative of the flow near the wall because they are large below the dune crest ($x = 0$), and also above the crest (Figure 4). A previous research by [54] also revealed that the peak values of the turbulent properties were located in the dune crest line from the channel bottom which was rarely affected by the bed roughness. Consequently, we selected the optimal roughness case from the results of the flow velocities in the $x$-$z$ plane, and $3D_{90}$ (=Case 3) proposed by [56] was chosen for the focus on the separation zone of the dune.

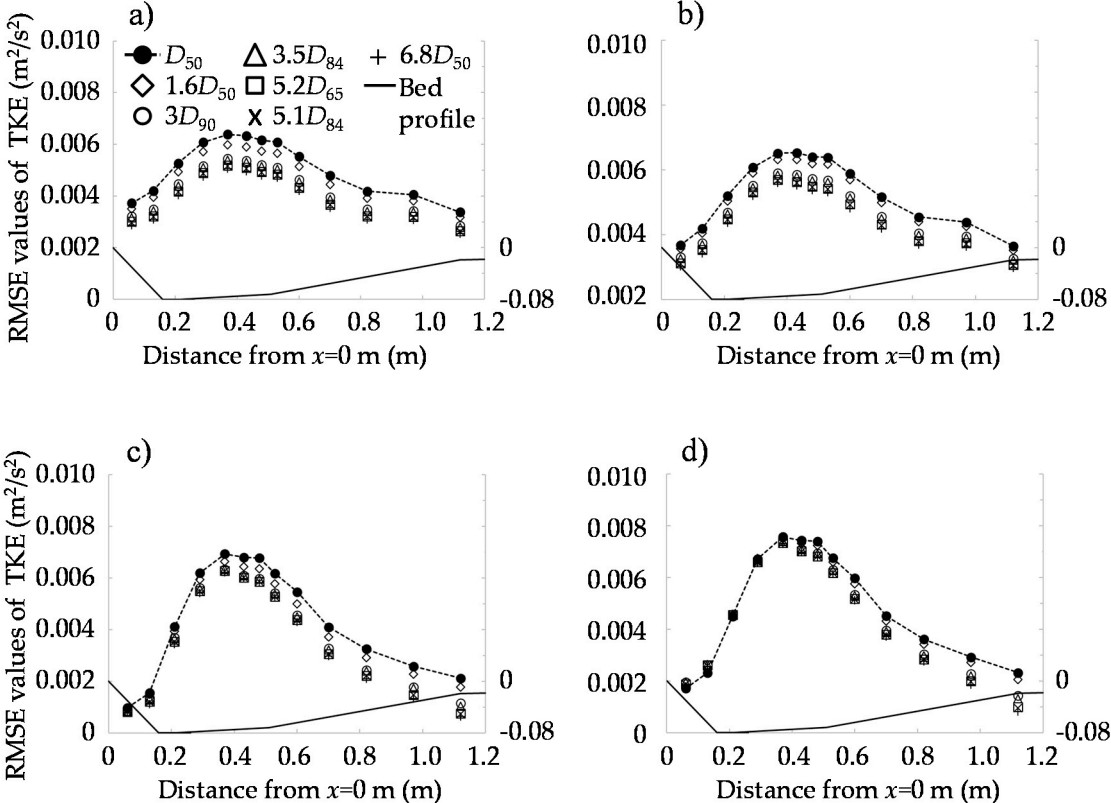

**Figure 8.** RMSE of TKE with various bottom roughness height formulas: (**a**) Fully depth-averaged ($0 < z/h_0 < 1$) RMSE with $k$-$\varepsilon$ model; (**b**) Fully depth-averaged with $k$-$\omega$ SST model; (**c**) Partially depth-averaged ($0 < z/h_0 < 0.15$) $k$-$\varepsilon$ model for the lower part; (**d**) Partially depth-averaged ($0 < z/h_0 < 0.15$) $k$-$\omega$ SST model.

For overall analysis, in Figure 9, flow velocity and turbulent kinetic energy data in Case 3 with $k$-$\varepsilon$ and $k$-$\omega$ SST models were compared with respect to the stream-wise location (Figure 9). In the figure, numerical results of the stream-wise flow velocity component were rather underestimated over

the entire section for all depth directions except for the lower part near the inlet where backwater occurred. Nevertheless, the depth-wise velocity showed a distribution that was somewhat consistent with the experimental results, except for several outliers in Figure 9c,d. In addition, a comparison of the TKE data showed that the numerical results were underestimated in Figure 9e,f. These results are considered to be a limitation of the RANS model application, and in the future, it can be improved by applying models such as LES. Nevertheless, the combination of the RANS model and roughness height found in this study secured applicability, because it was possible to adequately simulate the backwater generated in the lower part.

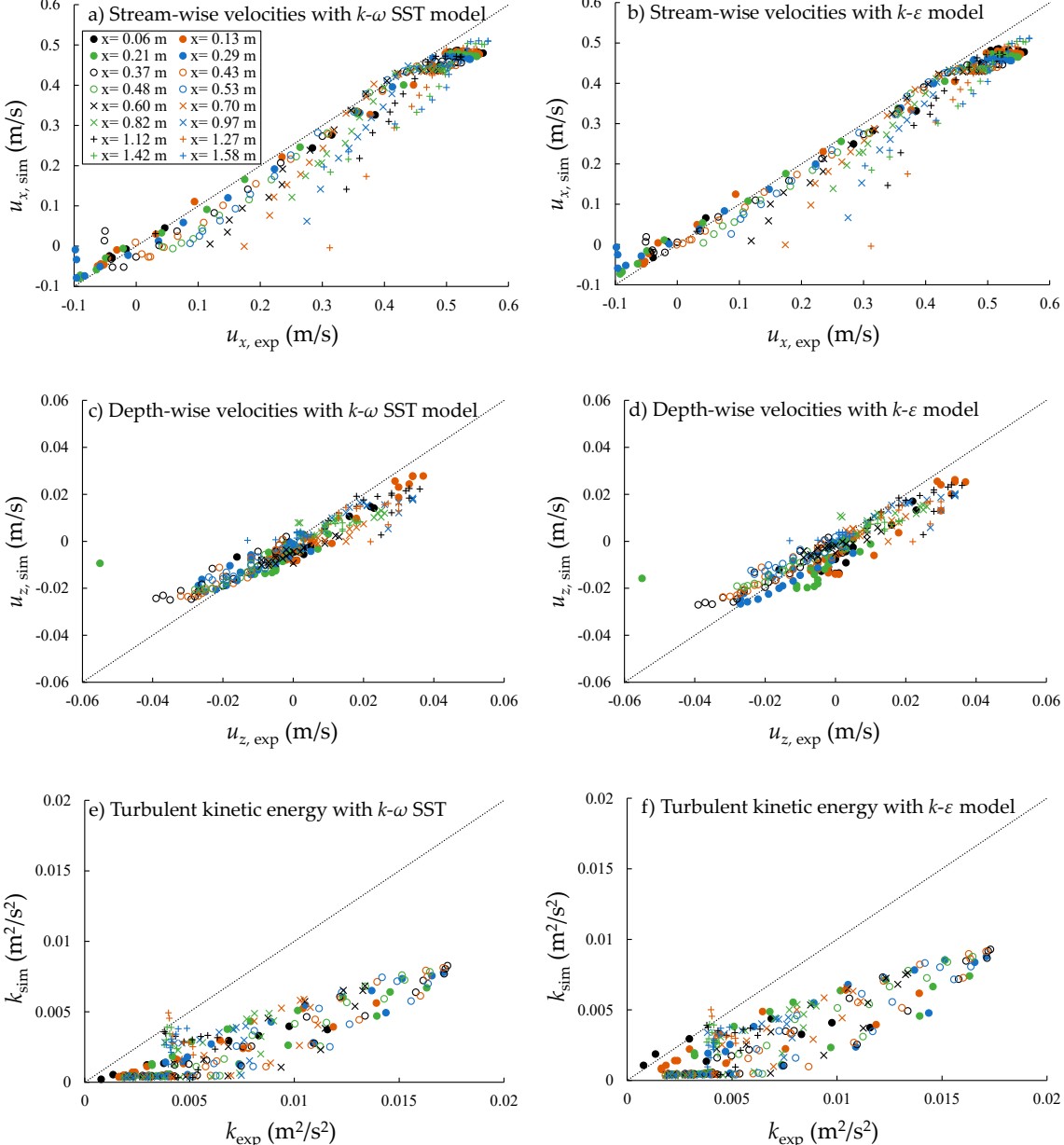

**Figure 9.** Comparison of simulated and experimental data of flow velocity and TKE with perfect line of agreement (= dashed line) of Case 3 ($k_s = 3D_{90}$).

## 5. Summary and Conclusions

This study focused on finding the appropriate numerical approach with roughness boundary formula at the channel bottom and turbulence scheme for modeling the flow and turbulence

characteristics in an artificial dune. Roughness values from seven formula for use in two turbulent models, $k$-$\varepsilon$ and $k$-$\omega$ SST, were compared. The simulated results of the flow velocity and TKE profiles were compared with the physical experimental data from [6], and the calculated results of 14 combined cases were also compared using the estimated RMSE values. The flow velocity profiles and TKE results from the $k$-$\varepsilon$ and $k$-$\omega$ SST models were calculated very similarly throughout the flow area in the artificial dune. Additionally, variations in roughness height affected the results of the turbulent modeling. Cases with higher roughness heights proposed previously were more accurate than those with lower values. However, in this study, we tried to find the appropriate combination of numerical modeling in the vicinity of the channel bottom. Therefore, we estimated the RMSE values in the separated zone for the lower and upper part from the initial water surface elevation and the overall part. In particular, the RMSE values of stream-wise velocity in the vertically lower part were remarkably lower than the fully depth-wise averaged RMSE values in the vicinity of the upstream part of the single dune. However, the overall RMSE values from the inlet to $x = 0.29$ m were the most accurate in Case 3. In terms of the depth-wise directional velocity, the results of the $k$-$\omega$ SST model upstream part were more accurate and the RMSE values in the wake region were smaller. Consequently, we determined that the combination of the $k$-$\omega$ SST model and Case 3 ($k_s = 3D_{90}$) was the most suitable for the research focus of the flow separation zone. Therefore, considering the accuracy of the whole study area, both turbulence models could be applied for modeling downstream of the separation zone with a wall function. On the basis of the combination suggested in this study, it is expected that more accurate combinations could be found in the future by supplementing additional turbulence analysis modules and adding behavior simulations in the width-direction geometry of the model.

**Author Contributions:** Conceptualization, J.A. and S.W.P.; methodology, J.L.; software, J.L.; validation, J.L., S.W.P., and J.A.; formal analysis, J.L.; investigation, J.L. and S.W.P.; resources, S.W.P.; data curation, J.L.; writing—original draft preparation, J.L.; writing—review and editing, J.A.; visualization, J.L. and S.W.P.; supervision, J.A.; project administration, J.A.; funding acquisition, J.A. All authors have read and agreed to the published version of the manuscript.

**Funding:** This research received no external funding.

**Acknowledgments:** This work was supported by Incheon National University Research Grant in 2018.

**Conflicts of Interest:** The authors declare no conflict of interest.

## Notations

| | |
|---|---|
| $\alpha_i, \beta_i, \sigma_i$ | Model closure coefficients ($\alpha_1 = 0.5532$, $\alpha_2 = 0.4403$, $\beta_1 = 0.075$, $\beta_2 = 0.0828$, $\sigma_k = 0.8503$, $\sigma_{\omega 1} = 0.5$, and $\sigma_{\omega 2} = 0.8561$) |
| $\beta^*$ | Empirical constant (=0.09) |
| $F_1, F_2$ | Blending functions |
| $a_1$ | Calibration coefficient (=0.31) |
| $CD_{k\omega}$ | Limited production function of the turbulent viscosity |
| $D_n$ | Diameter of particle intermediate axis for which $n = 50\%, 65\%, 84\%$, and $90\%$, respectively, of the sample of bed material is finer |
| $F_2$ | Second blending function |
| $h$ | Dune height |
| $h_0$ | Inlet water depth |
| $k$ | Turbulent kinetic energy |
| $k_s$ | Roughness height |
| $L$ | Dune length |
| $b$ | Dune width |
| $p$ | Pressure |
| $P_k$ | Turbulent viscosity production |
| $P_{\omega k}$ | Limited production term of the turbulent viscosity |
| $S$ | Absolute value of vorticity |
| $u_x$ | Flow velocities in stream-wise direction |

| | |
|---|---|
| $u_z$ | Flow velocities in water-depth direction |
| $u_0$ | Flow velocity at inlet |
| $u_{exp}$ | Observed values (flow velocity and turbulent kinetic energy) from experiments in the previous research |
| $u_{mod}$ | Calculated values (flow velocity and turbulent kinetic energy) from numerical modeling |
| $\vec{u}$ | Velocity vector |
| $x$ | Stream-wise direction |
| $z$ | Depth-wise directional distance |
| $t$ | time |
| $\Delta z$ | Mesh size of first node from the wall |
| $\varepsilon$ | Turbulent dissipation |
| $\nu$ | Kinematic viscosity coefficient |
| $\nu_T$ | Turbulent kinematic eddy viscosity coefficient |
| $\omega$ | Turbulent frequency (rate of dissipation of the eddies) |
| $T$ | Water temperature |

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
