# Peer review of "Optimal Strategy to Tackle a 2D Numerical Analysis of Non-Uniform Flow over Artificial Dune Regions: A Comparison with Bibliography Experimental Results"

_water, doi:10.3390/w12092331_

Round 1
Reviewer 1 Report
A file has been uploade, with comments.

Reviewer 2 Report
REVIEW:
Application of optimal combination of numerical analysis techniques in non-uniform flow over artificial dune regions
The manuscript deals with a numerical method for the simulation of flow over dunes. The research herein presented is certainly within the scope of Water.
According to my observations, the topic of the manuscript is interesting and challenging. However, the lack of clarity in some parts of the text should be addressed before the publication. I think the paper requires sharpening in the definition of the results obtained and subsequent discussion. Nonetheless, I am supportive with the manuscript and after the revision herein purposed I think it should be ready for publication. I will be happy to review an updated version of the manuscript.
List of comments:
- In the title I would remove the word “artificial”. It sounds weird while linking it to dunes.
- Introduction is well-written and sets an appropriate context. Nonetheless, the authors missed the following recent works [1,2] in line 52. I would advise authors to see those papers because they deal with LES numerical modelling and sediments in suspension, which are strongly connected with dunes formation and development.
- In equation 5, there is a typo: the empirical constant C3 is missing.
- Equations 3-5. Ok, this set of equations help to compute the turbulent viscosity, but please, provide a range of values. Are the authors sure that roughness is more relevant than turbulence to obtain accurate results?
- In the model setup, with a unique experimental validation, the mesh size is not specified. Furthermore, what happens if mesh is refined? There exists mesh convergence for this refinement level?. Lines 160 and 161 refer to Nguyen et al. but not to the present work.
- Section 4.1. Nothing is said in the text about the changing density of the water column due to the entrainment of suspended sediments. This points is relevant to accurately compute the velocity profiles, more specifically, closer to the dune surface where suspended sediment concentration is higher. This region is the one with larger error, as displayed in figures 2, 3, 4. See [3] for further details.
- Legend, symbols and axes are unreadable in figure 8. Please, magnify them.
References
[1] Large Eddy Simulations of sediment entrainment induced by a lock-exchange gravity current. Foteini et al. ADVANCES IN WATER RESOURCES. 2018.
[2] Drivers for mass and momentum exchange between the main channel and river bank lateral cavities. Ouro et al. ADVANCES IN WATER RESOURCES. 2020.
[3] One-Dimensional Riemann Solver Involving Variable Horizontal Density to Compute Unsteady Sediment Transport. Juez et al. JOURNAL OF HYDRAULIC ENGINEERING. 2015.
Round 2
Reviewer 1 Report
The notation is still to be improved. Several symbols in equations 1 to 12b are not included in the list. I found some α, β, several σ's and the coefficients of the turbulent model, at least. Therefore, the authors did not finished this part.
Final, conclusions should be better explained and improved.
Author Response
Dear reviewer,
Thank you for your valuable comments for our manuscript.
We tried to modify it as you mentioned.
thank you